# Benchmarking Bayesian Deep Learning on Diabetic Retinopathy Detection Tasks

**Neil Band**[*][†]
University of Oxford

**Tim G. J. Rudner**[*][†]
University of Oxford

**Qixuan Feng**
University of Oxford

**Angelos Filos**
University of Oxford

**Zachary Nado**
Google Research

**Michael W. Dusenberry**
Google Research

**Ghassen Jerfel**
Google Research

**Dustin Tran**
Google Research

**Yarin Gal**
University of Oxford

## Abstract

Bayesian deep learning seeks to equip deep neural networks with the ability to precisely quantify their predictive uncertainty, and has promised to make deep learning more reliable for safety-critical real-world applications. Yet, existing Bayesian deep learning methods fall short of this promise; new methods continue to be evaluated on unrealistic test beds that do not reflect the complexities of downstream real-world tasks that would benefit most from reliable uncertainty quantification. We propose the **RETINA Benchmark**, a set of real-world tasks that accurately reflect such complexities and are designed to assess the reliability of predictive models in safety-critical scenarios. Specifically, we curate two publicly available datasets of high-resolution human retina images exhibiting varying degrees of diabetic retinopathy, a medical condition that can lead to blindness, and use them to design a suite of automated diagnosis tasks that require reliable predictive uncertainty quantification. We use these tasks to benchmark well-established and state-of-the-art Bayesian deep learning methods on task-specific evaluation metrics. We provide an easy-to-use codebase for fast and easy benchmarking following reproducibility and software design principles. We provide implementations of all methods included in the benchmark as well as results computed over 100 TPU days, 20 GPU days, 400 hyperparameter configurations, and evaluation on at least 6 random seeds each.

## 1 Introduction

Bayesian deep learning has been applied successfully to a wide range of real-world prediction problems such as *medical diagnosis* [8, 28, 36, 65], *computer vision* [29, 30, 32], *scientific discovery* [37, 42], and *autonomous driving* [2, 17, 27, 30–32, 41].

Despite the demonstrated usefulness of Bayesian deep learning for such practical applications and a growing literature on inference methods [6, 14, 19, 21, 46, 47, 52, 61, 66], there exists no standardized benchmarking task that reflects the complexities and challenges of safety-critical real-world tasks while adequately accounting for the reliability of models' predictive uncertainty estimates.

To make meaningful progress in the development and successful deployment of reliable Bayesian deep learning methods, we need easy-to-use benchmarking tasks that reflect the real world and hence

---

[*]Equal contribution.
[†]Corresponding authors: neil.band@cs.ox.ac.uk and tim.rudner@cs.ox.ac.uk.

35th Conference on Neural Information Processing Systems (NeurIPS 2021) Track on Datasets and Benchmarks.

serve as a legitimate litmus test for practitioners that aim to deploy their models in safety-critical settings. Further, such tasks ought to be usable without the extensive domain expertise often necessary for appropriate experiment design and data preprocessing. Lastly, any such benchmarking task must include evaluation methods that test for predictive performance and assess different properties of models' predictive uncertainty estimates, while taking into account application-specific constraints.

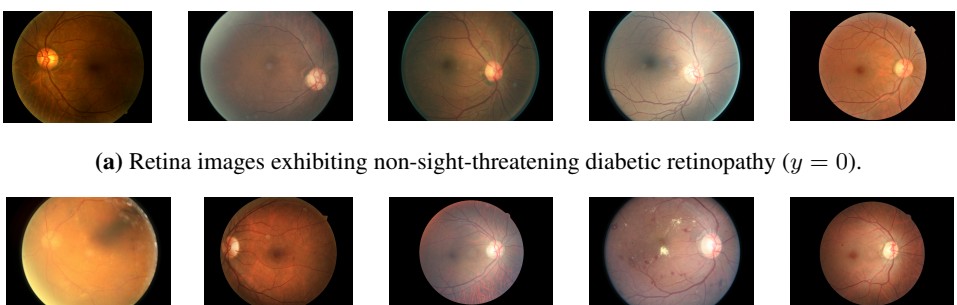

**(a)** Retina images exhibiting non-sight-threatening diabetic retinopathy ($y = 0$).

**(b)** Retina images exhibiting sight-threatening diabetic retinopathy ($y = 1$).

**Figure 1:** Samples of retina scans from the EyePACS dataset showing varying degrees of diabetic retinopathy.

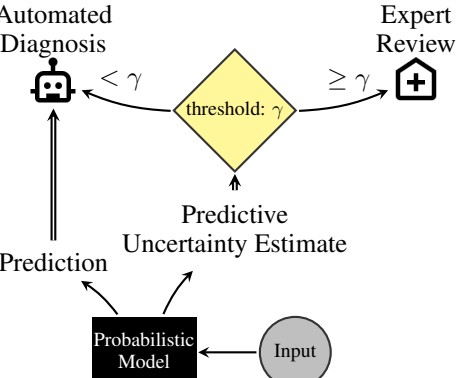

**Figure 2:** Automated diagnosis pipeline: For a given input, a model provides a prediction and a corresponding uncertainty estimate; if the uncertainty estimate is below a certain reference threshold $\gamma$ (indicating a low degree of uncertainty) the diagnosis is processed without further review; otherwise, it is referred to a medical expert.

In this paper, we propose a set of realistic safety-critical downstream tasks that respect these desiderata and use them to benchmark well-established and state-of-the-art Bayesian deep learning methods. To do so, we consider the problem of using machine learning to detect diabetic retinopathy, a medical condition considered the leading cause of vision impairment and blindness [54]. Unlike in prior works on diabetic retinopathy detection, the benchmarking tasks presented in this paper are specifically designed to assess the reliability of machine learning models and the quality of their predictive uncertainty estimates using both aleatoric and epistemic uncertainty estimates.

Medical diagnosis problems are particularly well-suited to assess reliability due to the severe harm caused by predictive models that make confident but poor predictions (for example, when a disease is not recognized). As a general desideratum, we want a model's predictive uncertainty to correlate with the correctness of its predictions. Good predictive uncertainty estimates can be a fail-safe against incorrect predictions. If a given data point might result in an incorrect prediction because it is meaningfully different from data in the training set—for example, because it shows signs of the disease not captured there, exhibits visual artifacts, or was obtained using different measurement devices—a good predictive model will express a high level of predictive uncertainty and flag the example for further review by a medical expert.

**Contributions.** We present the **RETINA Benchmark**: an easy-to-use, expert-guided, open-source *suite of diabetic retinopathy detection benchmarking tasks* for Bayesian deep learning. In particular, we design safety-critical downstream tasks from publicly available datasets. On these downstream tasks, we evaluate well-established and state-of-the-art Bayesian and non-Bayesian methods on a set of task-specific reliability and performance metrics. Lastly, we provide a modular and extensible implementation of the benchmarking tasks and methods, as well as pre-trained models obtained from an extensive hyperparameter optimization over more than 400 total configurations and evaluation, using over 100 TPU days and 20 GPU days of compute. Code to reproduce our results and benchmark new methods is available at:

github.com/google/uncertainty-baselines/.../diabetic_retinopathy_detection.

## 2 Downstream Benchmarking Tasks for Diabetic Retinopathy Detection

In this section, we present two real-world scenarios in diabetic retinopathy detection and describe how we merge two publicly available datasets to design corresponding prediction tasks.

### 2.1 Data and Preprocessing

**EyePACS Dataset.** We construct training datasets for different tasks from the EyePACS dataset, previously used for the Kaggle Diabetic Retinopathy Detection Challenge [13]. It contains high-resolution labeled images of human retinas exhibiting varying degrees of diabetic retinopathy. The dataset consists of 35,126 training, 10,906 validation, and 42,670 test images, each an RGB image of a human retina graded by a medical expert on the following scale: 0 (no diabetic retinopathy), 1 (mild diabetic retinopathy), 2 (moderate diabetic retinopathy), 3 (severe diabetic retinopathy), and 4 (proliferative diabetic retinopathy).

**APTOS Dataset.** To construct tasks that assess model performance under distribution shift, we use the APTOS 2019 Blindness Detection dataset [3]. The dataset also contains labeled images of human retinas exhibiting varying degrees of diabetic retinopathy, but was collected in India, from a different patient population, using different medical equipment. We use 80% of the images (2,929 images) as a test set and the other 20% (733 images) as a secondary validation set. Moreover, the images are significantly noisier than the images in the EyePACS dataset, with distinct visual artifacts (cf. Figure 7, Appendix A.8). Each image was graded on the same 0-to-4 scale as the EyePACS dataset.

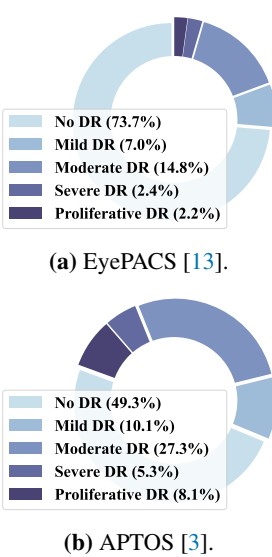

(a) EyePACS [13].

No DR (73.7%)
Mild DR (7.0%)
Moderate DR (14.8%)
Severe DR (2.4%)
Proliferative DR (2.2%)

(b) APTOS [3].

No DR (49.3%)
Mild DR (10.1%)
Moderate DR (27.3%)
Severe DR (5.3%)
Proliferative DR (8.1%)

**Figure 3:** Data class labels.

**Prediction Targets.** We follow Leibig et al. [36] and binarize all examples from both the EyePACS and APTOS datasets by dividing the classes up into sight-threatening diabetic retinopathy—defined as moderate diabetic retinopathy or worse (classes $\{2, 3, 4\}$)—and non-sight-threatening diabetic retinopathy—defined as no or mild diabetic retinopathy (classes $\{0, 1\}$). By international guidelines, this is the threshold at which a case should be referred to an ophthalmologist [64]. Example EyePACS retina images from the two classes are shown in Figure 1. Reflecting real-world challenges, the datasets are unbalanced—e.g., for EyePACS, only $19.6\%$ of the training set and $19.2\%$ of the test set have a positive label—and images have visual artifacts and noisy labels (some labels are incorrect).

**Data Preprocessing.** Data preprocessing on examples from both the EyePACS and APTOS datasets follows the winning entry of the Kaggle Challenge [13]: Images are rescaled such that retinas have a radius of 300 pixels, are smoothed using local Gaussian blur, and finally, are clipped to 90% size to remove boundary effects. Examples of original and corresponding processed images are provided in Figure 6 (Appendix A.8). We conduct an empirical study investigating how varying the strength of the Gaussian blur smoothing affects downstream performance and uncertainty quality in Appendix B.6.

### 2.2 Diabetic Retinopathy Detection under Severity Shift

Diabetes and diabetes-related illnesses such as diabetic retinopathy are becoming widespread. Yet cases of sight-threatening diabetic retinopathy are still relatively rare, and scans of retinas exhibiting signs of no or mild diabetic retinopathy are more easily obtainable. As a result, predictive models for detecting diabetic retinopathy may be trained on only a very small number of retina images showing signs of severe or proliferative retinopathy.

We design a prediction task that simulates this setting and allows us to assess the reliability of predictive models when they are evaluated on images that have been assigned a severity higher than any encountered in the training data. Specifically, we train models only on retina images showing signs of at most moderate diabetic retinopathy and evaluate them on retina images showing signs of severe or proliferative diabetic retinopathy. Given that many signs of moderate diabetic retinopathy are similar in appearance to signs of severe or proliferative diabetic retinopathy (just weaker), we would expect a good predictive model to be able to correctly classify the latter, but to exhibit increased predictive uncertainty. There are certain features of diabetic retinopathy progression that are unique to

more severe cases, such as vitreous hemorrhage, or bleeding into the vitreous humor [12]. However, we consider uncertainty-aware downstream tasks that tolerate such unfamiliar cases (cf. Section 2.4).

In this *Severity Shift* task, we partition the EyePACS dataset into a subset containing all retina images labeled as no, mild, or moderate diabetic retinopathy (original classes $\{0, 1, 2\}$) and a subset of retina images labeled as severe or proliferative diabetic retinopathy (original classes $\{3, 4\}$). Next, the samples in each subset are binarized (cf. Section 2.1): The subset of retina images showing signs of at most moderate diabetic retinopathy (subset "moderate") contains images of binarized classes $\{0, 1\}$; and the subset of retina images showing signs of severe or proliferative diabetic retinopathy (subset "severe") only contains the binarized class 1. This results in 33,545 images in the training set, and 40,727 and 3,524 images in the in-domain and distributionally shifted evaluation sets, respectively.

## 2.3 Diabetic Retinopathy Detection under Country Shift

Similar to the scarcity of scans of sight-threatening diabetic retinopathy, the availability of retina scans is limited in countries without widespread screening. Hence, a predictive model may be trained on images collected in the United States—where many scans are performed—and used to evaluate scans from another country, where scans are rarer and performed using different medical devices.

We design a prediction task that simulates this setting and allows us to evaluate the reliability of predictive models when the training and test data are not obtained from the same patient population nor collected with the same medical equipment. In this *Country Shift* task, we train models on retina images from the EyePACS dataset and evaluate them on retina images from the APTOS dataset. We use the entire training and test data provided in the EyePACS dataset and convert the task into binary classification as described in Section 2.1. This results in 35,126 images in the training set, and 42,670 and 2,929 images in the in-domain and distributionally shifted evaluation sets, respectively.

## 2.4 Downstream Task: Selective Prediction and Expert Referral

In real-world settings where the evaluation data may be sampled from a shifted distribution, incorrect predictions may become increasingly likely. To account for that possibility, predictive uncertainty estimates can be used to identify datapoints where the likelihood of an incorrect prediction is particularly high and refer them for further review as described in Figure 2. We consider a corresponding selective prediction task, where the predictive performance of a given model is evaluated for varying expert referral rates. That is, for a given referral rate of $\tau \in [0, 1]$, a model's predictive uncertainty is used to identify the $\tau$ proportion of images in the evaluation set for which the model's predictions are most uncertain. Those images are referred to a medical professional for further review, and the model is assessed on its predictions on the remaining $(1 - \tau)$ proportion of images. By repeating this process for all possible referral rates and assessing the model's predictive performance on the retained images, we estimate how reliable it would be in a safety-critical downstream task, where predictive uncertainty estimates are used in conjunction with human expertise to avoid harmful predictions.

Importantly, selective prediction tolerates out-of-distribution examples. For example, even if unfamiliar vitreous hemorrhages appear in certain *Severity Shift* images (cf. Section 2.2), a model with reliable uncertainty estimates will perform better in selective prediction by assigning these images high epistemic (and predictive) uncertainty, therefore referring them to an expert at a lower $\tau$. Appendix A.6 discusses best- and worst-case uncertainty estimates for the selective prediction task.

To assess how well different models' predictive uncertainty estimates can be used to separate correct from incorrect diagnoses, we perform selective prediction on three different evaluation settings for the prediction problems described in Sections 2.2 and 2.3, to account for the possibility that the evaluation dataset may contain samples from the in-domain distribution, a shifted distribution, or both.

## 2.5 Model Diagnostic: Predictive Uncertainty Histograms

We may also investigate how a model's predictive uncertainty estimates vary with respect to the ground-truth clinical label (0-to-4). For each task (*Country* or *Severity Shift*) and each uncertainty quantification method (cf. Section 5), we bin examples by their ground-truth clinical label. Then, for each (task, method, clinical label) tuple, we plot the distribution of predictive uncertainty estimates for correctly and incorrectly predicted examples (in blue and red, respectively). See Appendix B.1

for further setup details and plots for both tasks. A model that produces reliable uncertainty estimates should assign low predictive uncertainty to examples that it classifies correctly (the blue distribution should have most of its mass near $x = 0$) and high predictive uncertainty to examples that it classifies incorrectly (the red distribution should have its mass concentrated at a higher $x$-value).

## 3   Related Work

RETINA builds on prior works that demonstrated the usefulness of predictive uncertainty estimates in diabetic retinopathy detection and related downstream tasks [36]. We significantly extend the empirical evaluation in Leibig et al. [36] by designing new prediction problems and corresponding safety-critical downstream tasks for diabetic retinopathy detection, benchmarking a wide array of Bayesian deep learning methods, and providing a modular, extensible, and easy-to-use codebase. We also significantly extend Filos et al. [16] (of which this paper is a direct extension; with contributions from some of the authors), which does not consider severity shifts, only compares two variational inference methods, uses an outdated neural network architecture (with only ≈10% of the parameters of the ResNet-50 architecture used in this work), and considers only a small subset of the evaluation procedures included in RETINA (cf. Appendix B.4 for the full set of results).

Previous works have evaluated methods by predictive performance and quality of their predictive uncertainty estimates on curated datasets such as CIFAR-10 and FashionMNIST [47, 48, 24, 52]. Some prior works provide datasets and benchmarks for robustness and uncertainty quantification in real-world settings but have significant shortcomings. Le et al. [35] considers object detection using a real-world dataset [20] but benchmarks only two methods, neither of which can quantify epistemic uncertainty (cf. Section 4), and does not consider distribution shifts. Other works [5, 15] use methods which quantify both epistemic and aleatoric uncertainty, and consider distribution shifts, but use performance metrics which do not assess quality of uncertainty estimates, such as average precision and log-likelihood (cf. Section 6.3). Finally, Koh et al. [33] considers real-world datasets in domain adaptation problems, but restrictively assumes that the training data is composed of multiple training distributions with domain labels, and does not take into account models' predictive uncertainty.

In contrast, RETINA **(i)** considers real-world safety-critical tasks and accompanying uncertainty-aware metrics in an important application domain, **(ii)** is composed of large amounts of high-dimensional data (>80 GB), **(iii)** compares a larger set of methods than prior works and incorporates both aleatoric and epistemic uncertainty, and is implemented in adherence to the *Uncertainty Baselines* repository[3] practices for easy future use and extension, making it easier to benchmark other Bayesian deep learning methods not only on the tasks presented but also on a range of other datasets.

## 4   Uncertainty Estimation

Predictive models' total uncertainty can be decomposed into aleatoric and epistemic uncertainty. A model's aleatoric uncertainty is an estimate of the uncertainty inherent in the data (e.g., due to noisy inputs or targets), whereas a model's epistemic uncertainty is an estimate of the uncertainty due to constraints on the model (e.g., due to model misspecification) or the training process (e.g., due to convergence to bad local optima) [10]. Optimal uncertainty estimates would be perfectly correlated with the model error. Hence, because both aleatoric and epistemic uncertainty may contribute to an incorrect prediction, total uncertainty is our uncertainty measure of choice. For a model with stochastic parameters $\boldsymbol{\Theta}$, pre-likelihood outputs $f(\mathbf{X}; \boldsymbol{\Theta})$, and a likelihood function $p(\mathbf{y}_* \,|\, \mathbf{x}_*; \boldsymbol{\theta})$, the model's predictive uncertainty can be decomposed as

$$\underbrace{\mathcal{H}(\mathbb{E}[p(\mathbf{y}_* \,|\, f(\mathbf{x}_*; \boldsymbol{\theta}))])}_{\text{Total Uncertainty}} = \underbrace{\mathbb{E}[\mathcal{H}(p(\mathbf{y}_* \,|\, f(\mathbf{x}_*; \boldsymbol{\theta})))]}_{\text{Aleatoric Uncertainty}} + \underbrace{\mathcal{I}(\mathbf{y}_*; \boldsymbol{\Theta})}_{\text{Epistemic Uncertainty}}, \tag{1}$$

where the expectation is taken with respect to the distribution over the model parameters, $\mathcal{H}(\cdot)$ is the entropy functional, and $\mathcal{I}(\mathbf{y}_*; \boldsymbol{\Theta})$ is the mutual information between the model parameters and its predictions [9, 55].

---

[3]See https://github.com/google/uncertainty-baselines.

In binary classification settings with classes $\{0, 1\}$, the total predictive uncertainty is given by

$$\mathcal{H}(\mathbb{E}[p(\mathbf{y}_* \,|\, f(\mathbf{x}_*; \boldsymbol{\theta}))]) = -\sum_{c \in \{0,1\}} \mathbb{E}[p(\mathbf{y}_* = c \,|\, f(\mathbf{x}_*; \boldsymbol{\theta}))] \log \mathbb{E}[p(\mathbf{y}_* = c \,|\, f(\mathbf{x}_*; \boldsymbol{\theta}))], \quad (2)$$

where $f(\mathbf{x}_*; \boldsymbol{\theta})$ are logits and $p(\mathbf{y}_* = c \,|\, f(\mathbf{x}_*; \boldsymbol{\theta}))$ is a binary cross-entropy likelihood function. The total predictive uncertainty is high when either the aleatoric uncertainty is high (e.g., because the input is noisy), or when the epistemic uncertainty is high (e.g., because the model has many possible explanations for the input). In practice, the total predictive uncertainty $\mathcal{H}(\mathbb{E}[p(\mathbf{y}_* \,|\, f(\mathbf{x}_*; \boldsymbol{\theta}))])$ is computed with a Monte Carlo estimator $\mathbb{E}[p(\mathbf{y}_* \,|\, f(\mathbf{x}_*; \boldsymbol{\theta}))] \approx \frac{1}{S} \sum_i^S p(\mathbf{y}_* \,|\, f(\mathbf{x}_*; \boldsymbol{\theta}^{(i)}))$, where parameter realizations $\{\boldsymbol{\theta}^{(i)}\}_{i=1}^S$ are sampled from some distribution over the network parameters, and $p(\mathbf{y}_* \,|\, f(\mathbf{x}_*; \boldsymbol{\theta}^{(i)}))$ denotes the predictive distribution given parameter realization $\boldsymbol{\theta}^{(i)}$.

# 5 Methods

Estimating a model's predictive uncertainty in terms of both aleatoric and epistemic uncertainty requires a *distribution over predictive functions*. Such a distribution over predictive functions can be obtained by treating the parameters of a neural network as random variables and inferring a posterior distribution $p(\boldsymbol{\theta} \,|\, \mathcal{D})$—a distribution over the network parameters conditioned on a set of training data $\mathcal{D} = (\mathbf{X}_\mathcal{D}, \mathbf{y}_\mathcal{D})$—according to the rules of Bayesian inference. Neural networks with such distributions over the network parameters—referred to as a Bayesian neural networks (BNN)—induce distributions over functions that are able to capture both aleatoric and epistemic uncertainty [18, 38, 45]. Unfortunately, computing a posterior distribution over the parameters of a neural network according to the rule of Bayesian inference is analytically intractable and requires the use of approximate inference methods [18, 21, 25, 45, 50]. Below, we describe baseline and state-of-the-art methods for which we implemented standardized and optimized runscripts that are readily extensible for experimentation and deployment in application settings.

## 5.1 Maximum A Posteriori Estimation in Bayesian Neural Networks

As an alternative to inferring a posterior distribution over neural network parameters, maximum a posteriori (MAP) estimation yields network parameter values equal to the mode of the exact posterior distribution. For a prior distribution over network parameters with zero mean and precision $\lambda$, the maximum a posteriori estimate is equal to the solution of the $\ell_2$-regularized optimization problem $\arg\min_{\boldsymbol{\theta}}\{-\log p(\mathbf{y}_\mathcal{D} \,|\, f(\mathbf{X}; \boldsymbol{\theta})) + \lambda\|\boldsymbol{\theta}\|_2^2\}$, and as such is equivalent to parameter values obtained by training a neural network with weight decay. Since MAP estimation yields a point estimate of the MAP parameters, the MAP solution defines a deterministic neural network and is thus unable to capture any epistemic uncertainty. In classification tasks, they represent aleatoric uncertainty estimates via the predicted class probabilities [31]. We use neural networks with MAP estimation as a baseline for the benchmark.

## 5.2 Variational Inference in Bayesian Neural Networks

Variational inference is an approximate inference method that seeks to sidestep the intractability of exact posterior inference over the network parameters by framing posterior inference as a variational optimization problem. In particular, variational inference in neural networks seeks to find an approximation to the posterior distribution over parameters by solving the optimization problem

$$\mathrm{argmax}_{q \in \mathcal{Q}}\{\mathbb{E}_q[\log p(\mathbf{y}_\mathcal{D} \,|\, f(\mathbf{X}_\mathcal{D}; \boldsymbol{\theta}))] - \mathbb{D}_{\mathrm{KL}}(q \,\|\, p)\}, \quad (3)$$

where $\mathcal{Q}$ is a variational family of distributions and $p$ is a prior distribution.

**Gaussian Mean-Field Variational Inference.**   If $p \doteq p_{\boldsymbol{\Theta}}$ and $q \doteq q_{\boldsymbol{\Theta}}$ are distributions over parameters, $\mathcal{Q}$ is the family of mean-field (i.e., fully-factorized) Gaussian distributions, and the prior distribution over parameters $p_{\boldsymbol{\Theta}}$ is also a diagonal Gaussian, the resulting variational objective is amenable to stochastic variational inference and can be optimized using stochastic gradient methods [6, 21, 25, 26, 59]. Henceforth, we refer to BNN inference methods that make these variational assumptions as mean-field variational inference. To optimize this objective, the expectation is estimated using Monte Carlo sampling and the network parameters are reparameterized as $\boldsymbol{\Theta} \doteq \boldsymbol{\mu} + \boldsymbol{\sigma} \odot \boldsymbol{\epsilon}$ with $\boldsymbol{\epsilon} \sim \mathcal{N}(\mathbf{0}, \mathbf{I})$. Throughout, we use the flipout estimator [61] to reduce the variance of the gradient estimates, and temper the Kullback-Leibler divergence term in the variational objective [62].

**Radial-Gaussian Mean-Field Variational Inference.**    Radial-Gaussian mean-field variational inference [14] uses the same variational objective, prior, and variational distribution as standard Gaussian mean-field variational inference, but uses an alternative gradient estimator to obtain an improved signal-to-noise ratio in the gradient estimates. Specifically, the network parameters are reparameterized as $\boldsymbol{\Theta} \doteq \boldsymbol{\mu} + \boldsymbol{\sigma} \odot \frac{\boldsymbol{\epsilon}}{||\boldsymbol{\epsilon}||_2} \cdot |r|$ with $\boldsymbol{\epsilon} \sim \mathcal{N}(\mathbf{0}, \mathbf{I})$ and $r \sim \mathcal{N}(0, 1)$.

**Function-Space Variational Inference.**    Rudner et al. [52] proposed a tractable function-space variational objective for Bayesian neural networks. If $p \doteq p_{f([\mathbf{X}_\mathcal{D}, \mathbf{X}_\mathcal{I}]; \boldsymbol{\Theta})}$ and $q \doteq q_{f([\mathbf{X}_\mathcal{D}, \mathbf{X}_\mathcal{I}]; \boldsymbol{\Theta})}$ are distributions over functions evaluated at the training inputs $\mathbf{X}_\mathcal{D}$ and at a set of inducing inputs $\mathbf{X}_\mathcal{I}$, $\mathcal{Q}$ is the family of distributions over functions induced by some distribution over network parameters, and the Kullback-Leibler divergence between distributions over functions evaluated at $[\mathbf{X}_\mathcal{D}, \mathbf{X}_\mathcal{I}]$ is approximated by a linearization of the neural network mapping, then the resulting variational objective is amenable to stochastic variational inference [52, 53]. In RETINA, we define a Gaussian mean-field distribution over the final layer of the neural network and reparameterize the parameters as $\boldsymbol{\Theta} \doteq \boldsymbol{\mu} + \boldsymbol{\sigma} \odot \boldsymbol{\epsilon}$ with $\boldsymbol{\epsilon} \sim \mathcal{N}(\mathbf{0}, \mathbf{I})$.

**Monte Carlo Dropout.**    Gal and Ghahramani [19] showed that training a deterministic neural network with $\ell_2$- and dropout regularization [57], that is, solving the optimization problem $\arg\min_{\boldsymbol{\theta}}\{-\mathbb{E}_q[\log p(\mathbf{y}_\mathcal{D} \mid f(\mathbf{X}; \boldsymbol{\theta}))] + \lambda ||\boldsymbol{\theta}||_2^2\}$, where $q_{\boldsymbol{\Theta}}$ is the distribution over parameters obtained by applying dropout with a given dropout rate, approximately corresponds to variational inference in a Bayesian neural network. To sample from the approximate posterior predictive distribution, dropout is applied to the deterministic network parameters. To optimize the objective above, the expectation is estimated using a single Monte Carlo sample (i.e., by applying dropout).

**Rank-1 Parameterization.**    Dusenberry et al. [11] propose a rank-1 parameterization of Bayesian neural networks, where each weight matrix involves only a distribution on a rank-1 subspace, that is, each stochastic weight matrix is defined as $\mathbf{W}_k' = \mathbf{W}_k \odot \mathbf{r}_k \mathbf{s}_k^\top$, where $\mathbf{W}_k$ is a deterministic set of weights, and $\mathbf{r}_k$ and $\mathbf{s}_k$ are random vectors of parameters. Variational distributions over $\mathbf{r}_k$ and $\mathbf{s}_k$ and a Dirac delta distribution over $\mathbf{W}_k$ for all layers $k$ are obtained by optimizing a variational objective.

### 5.3   Model Ensembling

**Deep Ensembles.**    A deep ensemble [34] is a mixture of multiple independently-trained deterministic neural networks. As such, unlike BNNs, deep ensembles do not explicitly infer a distribution over the parameters of a single neural network. Instead, they marginalize over multiple deterministic models to obtain a predictive distribution that captures both aleatoric and epistemic uncertainty. We construct deep ensembles from multiple MAP neural networks trained with different random seeds.

**Ensembles of Bayesian Neural Networks.**    Ensembles of Bayesian neural networks [16, 52, 56] are mixtures of multiple independently-trained Bayesian neural networks. They can account for the possibility that any individual approximate posterior distribution obtained via variational inference may be a poor approximation to the exact posterior distribution and may hence yield a poor predictive distribution. A common issue in the Bayesian deep learning literature is that ensembles are frequently compared to single models, often due to computational constraints. In RETINA, we provide a unified comparison and construct ensembles for all predictive models, including BNNs.

## 6   RETINA Benchmark

### 6.1   Evaluation Protocol

**Network Architecture.**    We use a ResNet-50 architecture for all experiments [22]. A sigmoid transformation is applied to the final linear layer of all networks to obtain class probabilities corresponding to the outcomes of the binary classification problems described in Sections 2.2 and 2.3.

**Validation Data, Hyperparameter Tuning, and Monte Carlo Estimation.**    Reliable uncertainty estimation on data points from shifted distributions is the central challenge for Bayesian deep learning methods. In training and evaluating such methods, practitioners must decide how they should choose validation data: specifically, in which settings they would benefit from using "out-of-distribution" data points for hyperparameter tuning. We consider two real-world settings: **(i)** No distributionally shifted data is available during hyperparameter tuning. This setting reflects scenarios in which practitioners do not know what data or distributional shift they might encounter during deployment and hence cannot make assumptions about it at training time. **(ii)** Shifted validation data *is* available

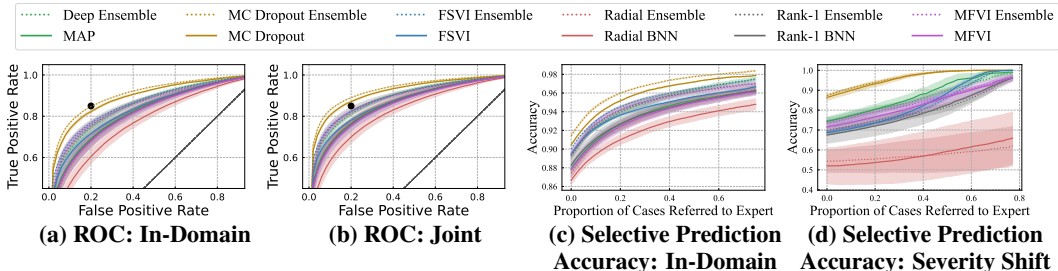

**Figure 4: Severity Shift.** We jointly assess model predictive performance and uncertainty quantification on the in-domain test dataset composed only of cases with either no, mild, or moderate diabetic retinopathy, and the *Severity Shift* evaluation set composed only of severe and proliferative cases. **Left:** The *receiver operating characteristic curve* (ROC) for (**a**) in-domain diagnosis and for (**b**) a joint dataset composed of examples from both the in-domain and *Severity Shift* evaluation sets. The dot in **black** denotes the NHS-recommended 85% sensitivity and 80% specificity ratios [63]. **Right:** Selective prediction on accuracy in the (**c**) in-domain and (**d**) *Severity Shift* settings. Shading denotes standard error computed over six random seeds. See Section 6.2.

for hyperparameter tuning. This setting reflects scenarios in which practitioners may intend to train a model on data collected from one subpopulation and deploy it on data collected from another subpopulation, but are able to acquire a small number of examples from the deployment subpopulation for use in tuning to improve generalization. Prior works on out-of-distribution detection [23] and uncertainty quantification [39] have considered setting (**ii**), but have not provided a comparative analysis, which would inform practitioners on when they ought to collect shifted validation data for tuning. We rigorously investigate the two settings across downstream tasks in Appendix B.4. In the main paper, we report results for models tuned under setting (**i**). Lastly, for all evaluations, we use five Monte Carlo samples per model to estimate predictive means (e.g., the MC DROPOUT ENSEMBLE with $K = 3$ ensemble members uses a total of $S = 15$ Monte Carlo samples).

The aim of the RETINA Benchmark is to adequately represent the challenges of real-world distributional shift, and rigorously assess the reliability of (Bayesian) uncertainty quantification in deep learning. Our selective prediction downstream tasks demonstrate two real-world use cases:

- **Tuning Referral Thresholds.** On the *Severity Shift* task, models demonstrate reasonable uncertainty estimates: predictive performance increases monotonically with an increasing referral rate $\tau$. Therefore, practitioners can infer which referral rate will lead to a desired predictive performance, or infer the performance for a predetermined referral rate (i.e., respecting a budget of expert time).

- **Detecting Low-Quality Predictive Uncertainty.** On the *Country Shift* task, most methods fail: predictive performance on the shifted dataset *declines* as $\tau$ increases, indicating that the quality of uncertainty estimates is no better than random referral. Importantly, this failure is *not reflected* in the standard performance measure for retinopathy diagnosis, the receiver operating characteristic (ROC) curve [36]—the area under the ROC curve (AUC) is *higher* on the shifted evaluation dataset than the in-domain dataset—meaning that a practitioner using only AUC might wrongly conclude that these models would perform well as part of an automated diagnosis pipeline (cf. Figure 2) on distributionally shifted data.

For each method, we assess both AUC and accuracy as a function of the referral rate $\tau$, evaluating the models' predictions for the $(1 - \tau)$ proportion of cases on which they are most certain, as indicated by their predictive uncertainty estimates. We additionally examine predictive uncertainty histograms for each task, method, and ground-truth clinical label (cf. Section 2.5, Appendix B.1) to determine if methods have particularly good or bad uncertainty estimates at particular severity levels. We also investigate other metrics to assess the reliability of models' uncertainty estimates, including *expected calibration error* and *out-of-distribution detection AUC*, in Appendix B.4.

## 6.2 Severity Shift

On the *Severity Shift* task (Figure 4, Table 2), models are trained on EyePACS images that show signs of at most moderate diabetic retinopathy. We assess their ability to generalize to images showing signs of severe or proliferative retinopathy. Surprisingly, we find that models generalize well from cases with no worse than moderate diabetic retinopathy (in-domain) (Figure 4(**a**)) to severe cases (Figure 4(**b**)), improving their AUC under the distribution shift.

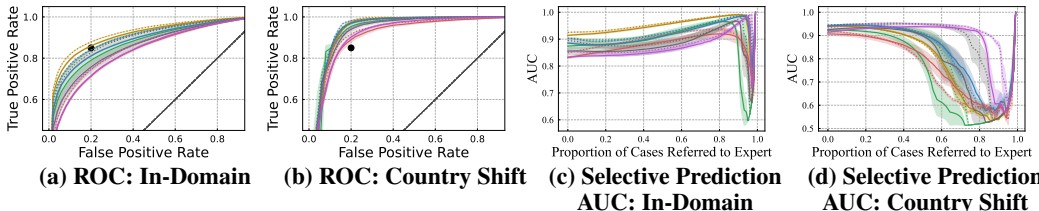

**(a) ROC: In-Domain**  **(b) ROC: Country Shift**  **(c) Selective Prediction AUC: In-Domain**  **(d) Selective Prediction AUC: Country Shift**

**Figure 5: Country Shift.** We jointly assess model predictive performance and uncertainty quantification on both in-domain and distributionally shifted data. **Left:** The *receiver operating characteristic curve* (ROC) for in-population diagnosis on the (**a**) EyePACS [13] test set and for (**b**) changing medical equipment and patient populations on the APTOS [3] test set. The dot in **black** denotes the NHS-recommended 85% sensitivity and 80% specificity ratios [63]. **Right:** *selective prediction* on AUC in the (**c**) EyePACS [13] and the (**d**) APTOS [3] settings. Shading denotes standard error computed over six random seeds. See Section 6.3.

**Methods Generalize Reasonably Well Under Severity Shift.**     Reliable predictive uncertainty estimates correlate with predictive error, and therefore we would expect a model's performance (e.g., measured in terms of accuracy or AUC) to increase as more examples on which the model exhibits high uncertainty are referred to an expert. On both the in-domain and *Severity Shift* evaluation sets (Figures 4(**c**) and (**d**)), models demonstrate reasonable uncertainty in that accuracy monotonically increases as $\tau$ increases. This highlights two ways that practitioners may use selective prediction to prepare models for a real-world deployment in the presence of potential distribution shifts. First, given a performance target (e.g., $\geq 95\%$ accuracy) the referral curve can be used to determine the minimum $\tau$ achieving this target, estimating a medical experts' workload. Second, for a maximum acceptable referral rate (e.g., a clinic has medical experts to handle referral of $\tau \leq 20\%$ of patients) the referral curve can be used to determine the optimal $\tau$ value and the corresponding performance. For monotonically increasing referral curves, the optimal $\tau$ is uniquely the maximum acceptable referral rate.

**Taking into Account Epistemic Uncertainty Can Improve Reliability.**     On the *Severity Shift* task (Figure 4(**d**)) many models achieve near-perfect accuracy well before all examples have been referred. For example, MC DROPOUT, which incorporates both epistemic and aleatoric uncertainty (cf. Section 4), achieves 100% predictive accuracy near the 50% referral rate—nearly 20% lower than the referral rate at which a deterministic neural network (MAP), which only represents aleatoric uncertainty, achieves this level of accuracy. Other variational inference methods underperform MAP, underscoring the importance of continued work on approximate inference in BNNs.

**Predictive Uncertainty Histograms Identify Harmful Uncertainty Quantification.**     In Figure 8 (Appendix B.1), we find that MAP, RANK-1, and MFVI generate worse uncertainty estimates than other methods on the shifted data (labels 3 and 4); many of their incorrect predictions are assigned low predictive uncertainty (i.e., the red distribution is concentrated near 0). These include false negatives with low uncertainty which are particularly dangerous in automated diagnosis settings (cf. Figure 2), as a medical expert would not be able to catch the model's failure to recognize the condition.

### 6.3    Country Shift

In the *Country Shift* task (Figure 5, Table 1), we consider the performance of models trained on the US EyePACS [13] dataset and evaluated under distributional shift, on the Indian APTOS dataset [3]. The left two plots of Figure 5 present the ROC curves of methods evaluated on the in-domain (**a**) and *Country Shift* (**b**) evaluation datasets. The black dot in Figures 5(**a**) and (**b**) denotes the minimum sensitivity–specificity threshold for the deployment of automated diabetic retinopathy diagnosis systems set by the British National Health Service (NHS) [63]. On the in-domain test dataset, only the MC DROPOUT variants meet the NHS standard; on the APTOS dataset, essentially all methods surpass the standard.[4] Hence, practitioners using only the ROC curve and its AUC (cf. Table 1) might conclude that their model generalizes under the distribution shift although the ROC curve provides no information on the application of uncertainty estimates to real-world scenarios (cf. Figure 2).

**Selective Prediction Can Indicate Failures in Uncertainty Estimation.**     Unlike the ROC curve, the selective prediction metric conveys how a model would perform in an automated diagnosis

---

[4]We investigate this in Appendix B.5 and find that class proportions do not account for the improved predictive performance on APTOS, implying other contributing factors such as demographics or camera type.

**Table 1: Country Shift.** Prediction and uncertainty quality of baseline methods in terms of the area under the receiver operating characteristic curve (AUC) and classification accuracy, as a function of the proportion of data referred to a medical expert. All methods are tuned on in-domain validation AUC, and ensembles have $K = 3$ constituent models (true for all subsequent tables unless specified otherwise). On in-domain data, MC DROPOUT performs best across all thresholds. On distributionally shifted data, no method consistently performs best.

| Method | No Referral | | 50% Data Referred | | 70% Data Referred | |
|---|---|---|---|---|---|---|
| | AUC (%) ↑ | Accuracy (%) ↑ | AUC (%) ↑ | Accuracy (%) ↑ | AUC (%) ↑ | Accuracy ↑ |
| EyePACS Dataset (In-Domain) | | | | | | |
| MAP (Deterministic) | $87.4_{\pm1.3}$ | $88.6_{\pm0.7}$ | $91.1_{\pm1.8}$ | $95.9_{\pm0.4}$ | $94.9_{\pm1.1}$ | $96.5_{\pm0.3}$ |
| MFVI | $83.3_{\pm0.2}$ | $85.7_{\pm0.1}$ | $85.5_{\pm0.7}$ | $94.5_{\pm0.1}$ | $88.2_{\pm0.7}$ | $95.9_{\pm0.1}$ |
| RADIAL-MFVI | $83.2_{\pm0.5}$ | $74.2_{\pm5.0}$ | $88.9_{\pm0.9}$ | $81.8_{\pm6.0}$ | $91.2_{\pm1.3}$ | $83.8_{\pm5.5}$ |
| FSVI | $88.5_{\pm0.1}$ | $89.8_{\pm0.0}$ | $91.0_{\pm0.4}$ | $96.4_{\pm0.0}$ | $94.3_{\pm0.3}$ | $97.2_{\pm0.1}$ |
| MC DROPOUT | $91.4_{\pm0.2}$ | $90.9_{\pm0.1}$ | $95.3_{\pm0.2}$ | $97.4_{\pm0.1}$ | $97.4_{\pm0.1}$ | $98.1_{\pm0.0}$ |
| RANK-1 | $85.6_{\pm1.4}$ | $87.7_{\pm0.8}$ | $87.1_{\pm2.3}$ | $95.3_{\pm0.5}$ | $90.9_{\pm2.0}$ | $96.4_{\pm0.4}$ |
| DEEP ENSEMBLE | $90.3_{\pm0.2}$ | $90.3_{\pm0.3}$ | $91.7_{\pm0.6}$ | $97.2_{\pm0.0}$ | $95.0_{\pm0.5}$ | $97.9_{\pm0.0}$ |
| MFVI ENSEMBLE | $85.4_{\pm0.0}$ | $87.8_{\pm0.0}$ | $86.3_{\pm0.4}$ | $95.4_{\pm0.0}$ | $89.2_{\pm0.4}$ | $96.7_{\pm0.1}$ |
| RADIAL-MFVI ENSEMBLE | $84.9_{\pm0.1}$ | $74.2_{\pm1.5}$ | $91.4_{\pm0.2}$ | $83.4_{\pm1.7}$ | $93.3_{\pm0.3}$ | $85.9_{\pm1.6}$ |
| FSVI ENSEMBLE | $90.3_{\pm0.1}$ | $90.6_{\pm0.0}$ | $92.1_{\pm0.2}$ | $97.1_{\pm0.0}$ | $95.2_{\pm0.2}$ | $97.8_{\pm0.1}$ |
| MC DROPOUT ENSEMBLE | $\mathbf{92.5_{\pm0.0}}$ | $\mathbf{91.6_{\pm0.0}}$ | $\mathbf{95.8_{\pm0.1}}$ | $\mathbf{97.8_{\pm0.0}}$ | $\mathbf{97.7_{\pm0.1}}$ | $\mathbf{98.4_{\pm0.0}}$ |
| RANK-1 ENSEMBLE | $89.5_{\pm0.8}$ | $89.3_{\pm0.4}$ | $88.5_{\pm1.3}$ | $96.9_{\pm0.3}$ | $91.6_{\pm1.2}$ | $97.6_{\pm0.3}$ |
| APTOS 2019 Dataset (Population Shift) | | | | | | |
| MAP (Deterministic) | $92.2_{\pm0.2}$ | $86.2_{\pm0.6}$ | $80.1_{\pm3.6}$ | $87.6_{\pm1.5}$ | $55.4_{\pm4.3}$ | $85.4_{\pm1.2}$ |
| MFVI | $91.4_{\pm0.2}$ | $84.1_{\pm0.3}$ | $93.8_{\pm0.4}$ | $92.1_{\pm0.5}$ | $93.0_{\pm0.6}$ | $92.7_{\pm0.5}$ |
| RADIAL-MFVI | $90.7_{\pm0.7}$ | $71.8_{\pm4.6}$ | $82.0_{\pm2.5}$ | $81.5_{\pm2.7}$ | $66.4_{\pm2.1}$ | $85.9_{\pm1.0}$ |
| FSVI | $94.1_{\pm0.1}$ | $87.6_{\pm0.5}$ | $90.6_{\pm0.9}$ | $90.7_{\pm0.7}$ | $77.2_{\pm4.6}$ | $89.8_{\pm0.3}$ |
| MC DROPOUT | $94.0_{\pm0.2}$ | $86.8_{\pm0.2}$ | $87.4_{\pm0.3}$ | $88.1_{\pm0.2}$ | $65.3_{\pm1.7}$ | $88.2_{\pm0.4}$ |
| RANK-1 | $92.5_{\pm0.3}$ | $86.2_{\pm0.5}$ | $90.1_{\pm2.5}$ | $91.4_{\pm1.1}$ | $75.1_{\pm7.8}$ | $89.5_{\pm1.5}$ |
| DEEP ENSEMBLE | $94.2_{\pm0.2}$ | $87.5_{\pm0.1}$ | $91.2_{\pm1.9}$ | $92.4_{\pm0.9}$ | $67.4_{\pm7.3}$ | $90.1_{\pm1.2}$ |
| MFVI ENSEMBLE | $93.2_{\pm0.1}$ | $87.0_{\pm0.2}$ | $\mathbf{94.9_{\pm0.3}}$ | $\mathbf{93.7_{\pm0.3}}$ | $\mathbf{94.2_{\pm0.3}}$ | $\mathbf{94.0_{\pm0.3}}$ |
| RADIAL-MFVI ENSEMBLE | $91.8_{\pm0.2}$ | $69.0_{\pm1.9}$ | $78.6_{\pm0.6}$ | $79.8_{\pm0.9}$ | $60.9_{\pm0.3}$ | $86.7_{\pm0.2}$ |
| FSVI ENSEMBLE | $\mathbf{94.6_{\pm0.1}}$ | $\mathbf{88.9_{\pm0.2}}$ | $90.7_{\pm0.5}$ | $91.1_{\pm0.6}$ | $74.1_{\pm3.4}$ | $89.8_{\pm0.2}$ |
| MC DROPOUT ENSEMBLE | $94.1_{\pm0.1}$ | $87.6_{\pm0.1}$ | $86.8_{\pm0.2}$ | $88.0_{\pm0.2}$ | $62.3_{\pm0.4}$ | $87.7_{\pm0.2}$ |
| RANK-1 ENSEMBLE | $94.1_{\pm0.2}$ | $88.3_{\pm0.2}$ | $\mathbf{94.9_{\pm0.4}}$ | $93.5_{\pm0.3}$ | $92.4_{\pm1.5}$ | $93.8_{\pm0.3}$ |

pipeline in which the reliability of models' uncertainty estimates directly impacts performance (cf. Figure 2). Recall that if a model generates reliable predictive uncertainty estimates, the AUC should increase as more patients with uncertain predictions are referred for expert review. This mechanism is illustrated well by the application of MFVI to the *Country Shift* task (Figure 5(**d**) and Table 1), since the AUC improves from an initial $91.4\%$ up to $93.8\%$ when referring 50% of the patients, but then deteriorates as the model is forced to refer patients on which it is both certain and correct. In contrast, other models' AUCs trend downwards; using uncertainty to refer patients actively hurts model performance on this shifted dataset.

**Different Prediction Tasks Yield Different Method Rankings.** In Figure 5(**c**), variational inference methods, including MC DROPOUT, FSVI, and DEEP ENSEMBLE, outperform MAP inference. This highlights that rankings are task-dependent, and underscores the importance of generic evaluation frameworks to enable rapid benchmarking on many tasks.

## 7 Conclusions

The deployment of modern machine learning models in safety-critical real-world settings necessitates trust in the reliability of the models' predictions.

To encourage the development of Bayesian deep learning methods that are capable of generating reliable uncertainty estimates about their predictions, we introduced the RETINA Benchmark, a set of safety-critical real-world clinical prediction tasks which highlight various shortcomings of existing uncertainty quantification methods. We demonstrate that by taking into account the quality of predictive uncertainty estimates, selective prediction can help identify whether methods might fail when deployed as part of an automated diagnosis pipeline (cf. Figure 2), whereas standard metrics such as ROC curves cannot.

While no single set of benchmarking tasks is a panacea, we hope that the tasks and evaluation methods presented in RETINA will significantly lower the barrier for assessing the reliability of Bayesian deep learning methods on safety-critical real-world prediction tasks.

## Acknowledgments and Disclosure of Funding

We thank Google Research for providing computational and storage resources. We thank Intel Labs for their computational support. We thank Ranganath Krishnan for his contributions to the RANK-1 implementation by porting a CIFAR-10 training script, and for sharing his expertise on variational inference in Bayesian neural networks. We thank Sebastian Farquhar for his contributions to the RADIAL-MFVI implementation, including a TensorFlow Probability distribution, code review, and feedback on tied means, $\ell_2$-regularization, variance reduction, and hyperparameters. We thank Jorge Cuadros, OD, PhD (CEO of EyePACS) and Jan Brauner, MD (University of Oxford) for lending their domain expertise in task design. We thank all other contributors to the *Uncertainty Baselines* project [43] (into which this benchmark is integrated): Mark Collier, Josip Djolonga, Marton Havasi, Rodolphe Jenatton, Jeremiah Liu, Zelda Mariet, Jeremy Nixon, Shreyas Padhy, Jie Ren, Faris Sbahi, Yeming Wen, Florian Wenzel, Kevin Murphy, D. Sculley, Balaji Lakshminarayanan, and Jasper Snoek. NB and TGJR acknowledge funding from the Rhodes Trust. TGJR also acknowledges funding from Qualcomm and the Engineering and Physical Sciences Research Council (EPSRC).

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
