# OpenReview forum: "Benchmarking Bayesian Deep Learning on Diabetic Retinopathy Detection Tasks"
_NeurIPS.cc/2021/Track/Datasets_and_Benchmarks/Round2 — NeurIPS 2021 Datasets and Benchmarks Track (Round 2)_

### Official Review · Reviewer_9xr4 · 2021-09-19
**Thorough review of Bayesian methods for uncertainty**

**Rating:** 7
**Confidence:** 4
**Correctness:** To my understanding, the evaluation m…
**Clarity:** The paper is well structured and the …

**Strengths:**

*The paper is well structured and written using a clear language.
* Both tasks are similar to what a real-life predictive model may encounter, depending on its training data and intended usage.
* A wide variety of uncertainty methods are benchmarked on the two proposed tasks.
* All data and code used in this paper is publicly available.


**Weaknesses:**

* For a more thorough benchmarking process, the addition of a more modern network architecture (e.g. EfficientNet) could be beneficial.
* Estimated Calibration Error (ECE) is a commonly reported metric in uncertainty literature. Adding it would give readers a better understanding of the models’ performance and would make future comparisons easier.


**Additional Feedback:**

-

**Documentation:**

A link to the code used in this paper is provided.

**Ethics:**

No concerns come to mind.

**Relation To Prior Work:**

Prior work is discussed with a focus on how this paper improves over the previous work from the same authors.

**Summary And Contributions:**

The authors introduce an uncertainty benchmark for diabetic retinopathy detection. They propose two tasks, both focused on out-of-distribution data. The first task trains on a dataset and tests on another, while the second task trains on mild to moderate cases and tests on severe cases of diabetic retinopathy. A multitude of uncertainty methods are benchmarked on these two tasks.

---

> ### Author Response · Authors · 2021-09-29
> **Response to Reviewer 9xr4**
>
> Dear Reviewer 9xr4,
>
> Thanks for your feedback. We address general comments in the reply above. Specific comments are addressed below.
>
> --
>
> ### Expected Calibration Error
>
> Thank you for the suggestion!
> We agree that inclusion of expected calibration error is important to better understand model performance and make comparisons to future work.
> We do in fact include experimental results on ECE across all tasks, uncertainty quantification methods, both tuning regimes (using in-domain only, and both in-domain and shifted data), and all splits of evaluation data (in-domain, shifted, in-domain + shifted) in Appendix D.
> In order to avoid any confusion, we now explicitly state in Section 6.1 (Evaluation Protocol) that we conduct experiments using ECE and direct the reader to the respective results in the Appendix.
> Similarly, we mention another common metric in uncertainty literature which we also evaluate: out-of-distribution detection AUC.
>
>
> ### EfficientNet and Other Network Architectures
>
> We are investigating the performance of a state-of-the-art computer vision architecture---with over 50M parameters, pretrained on 10M+ images---for a subset of approximate inference methods, on our benchmark. We plan to release corresponding model checkpoints on the GitHub repository.
>
> --
>
> We hope that our response and the new experiments have addressed your questions and concerns. Please let us know if you have any remaining questions.

---

### Official Review · Reviewer_LzQ9 · 2021-09-19
**This paper is the first present a suite of tasks for benchmarking Bayesian deep learning techniques in the context of diabetic retinopathy detection.**

**Rating:** 7
**Confidence:** 3
**Clarity:** The paper is well-written, and easy t…

**Strengths:**

To my knowledge, there is no other dataset or benchmark that specifically aims to evaluate performance of Bayesian deep learning approaches. This is the major strength of this paper, and will be appreciated by the broader community. The paper also has important social implications, since it aims to evaluate how a medical diagnosis system would perform in a real-world clinical setting.


**Weaknesses:**

One thing that is missing from the documentation is the expected format of the data. I realize that this paper is about a benchmark, and not a dataset. However, someone that reads this paper may want to take their own retinopathy dataset and apply the authors’ code. In this case, there is no documentation about how to structure the dataset in a way that is understandable by the authors’ codebase.

**Additional Feedback:**

-	I see that the APTOS 2019 dataset was graded by clinicians using the same 0-4 scale as the EyePACS dataset. You may want to mention this in the description of the APTOS dataset
-	Keep the symbol for “referral rate” consistent. You define it as “X” in Section 2.4, then redefine it as “\tau” in Section 2.6. Also note that “\tau” is already used to denote the precision in the prior distribution over the network parameters in Section 5.1.
-	In Figure 6 (supplementary information) – are these images only from the EyePACS dataset? Or do these include APTOS images as well? It will be helpful to the reader to make that distinction. Also, including example images from both datasets may help the reader further understand the difficulty of the Country Shift prediction task.


**Correctness:**

The authors correctly state in the title, and throughout the paper, that they are proposing a benchmark (not a dataset). The dataset described in the paper consists of two publicly released datasets which have been curated by the authors to form the two selective prediction tasks. The experiment design makes plenty of sense for benchmarking Bayesian deep learning approaches.

**Documentation:**

The URL provides a link to the authors’ Github repository containing code, implementations and model checkpoints of the Bayesian deep learning approaches evaluated in the paper, and documentation about how to run a pre-trained model through a selective prediction task.

**Ethics:**

None that I know of.

**Relation To Prior Work:**

It is clearly discussed how this work differs from previous contributions, specifically Refs [13] and [29].

**Summary And Contributions:**

This paper presents a suite of tasks and pre-trained models for benchmarking Bayesian deep learning techniques in the context of diabetic retinopathy detection. Two pre-existing datasets (EyePACS and APTOS 2019) containing high-resolution images of human retinas are curated to develop benchmarks in which models are trained, and then evaluated on distributionally shifted data. The authors propose two tracks in which realistic distribution shifts are applied to the evaluation data: 1) images acquired in a different country than the training data (Country Shift) and 2) images with higher severity of diabetic retinopathy than the training data (Severity Shift). For each track, a selective prediction task is performed in which the uncertainty estimated by the Bayesian neural network is used to refer images to a specialist if the network is unsure about its prediction. The authors demonstrate that their proposed tasks are more informative about real-world model performance than standard metrics, such as prediction accuracy and receiver-operator characteristic curve analysis.

---

> ### Author Response · Authors · 2021-09-29
> **Response to Reviewer LzQ9**
>
> Dear Reviewer LzQ9,
>
> Thanks for your feedback. We address general comments in the reply above. Specific comments are addressed below.
>
> --
>
> ### Expected Data Format, Extensibility to Other Retinopathy Datasets
>
> Thank you for the suggestion. The implementation of the APTOS dataset is an example of how to structure an arbitrary vision dataset to work with our framework---we load directly from raw images and a metadata CSV, and apply the preprocessing that was originally designed for the EyePACS data. We have updated the discussion of benchmark extensibility (Appendix A) to explain this and direct practitioners to the [relevant part](https://github.com/google/uncertainty-baselines/tree/main/uncertainty_baselines/datasets) of the GitHub repository. The APTOS dataset will be added there shortly.
>
>
> ### APTOS Grading Scale
>
> This is an important clarification as the overlapping labels between EyePACS and APTOS allows us to investigate robustness on the APTOS dataset, as opposed to other setups in uncertainty quantification with disjoint label sets between the in-domain and out-of-domain datasets (e.g., open-set recognition). We have updated Section 2.1 to mention this.
>
>
> ### Referral Rate Notation
>
> Thanks for drawing our attention to this notational inconsistency. We have switched the precision of the prior distribution over the network parameters (the weight decay coefficient) to $\lambda$ following standard notation, and have switched the references to the referral rate in Section 2.4 to use $\tau$ instead of “X”.
>
>
> ### EyePACS and APTOS Image Examples
>
> The images in Figure 6 are indeed only from the EyePACS dataset. We agree that it would be helpful to include example images from the APTOS dataset, and have done so in an additional figure (Figure 7, Appendix B). The figure illustrates some of the challenges of that dataset (which is collected with cheaper collection devices) including blur, low-lighting conditions, and artifacts around the edge of the retina.
>
> --
>
> We hope that our response and the new experiments have addressed your questions and concerns. Please let us know if you have any remaining questions.

---

### Official Review · Reviewer_7w45 · 2021-09-19
**Good paper, may require additional details and experimentation for benchmarking tasks and uncertainty quantification**

**Rating:** 7
**Confidence:** 3
**Correctness:** The claims made are correct and bench…
**Clarity:** The paper is well written

**Strengths:**

1.	The work tackles an important problem of contextualizing model performance based on model uncertainty, which is critical for safety-critical applications.
2.	The code release appears to be quite polished and should help simplify user interaction and future experimentation of these methods


**Weaknesses:**

1.	The work mentions that no existing dataset benchmarks exist for real-world tasks, but there are several datasets and analysis done for other safety critical, real-world tasks such as the KITTI dataset for autonomous driving [1, 2].
2.	While the work makes a point of emphasizing the importance of uncertainty quantification, details regarding how the uncertainty metric is quantified is unclear in the main text and appendix. This makes it difficult to follow how these metrics are calculated for the different methods, and how this calculation impacts the selective prediction evaluation.
3.	The work also converts a multi-class problem into a single class problem. While this may be commonly done for previous DR challenges, this work can benefit largely from exploring the dynamics of uncertainty in this multi-class setting where the classes are grades of severity of DR. Taking the Severity Task as an example, it would be useful to quantify how uncertainty for predictions is correlated with the non-binarized severity of the disease. This may provide more insight into the observation that there was minimal loss in performance when evaluating on the OOD test set in the Severity Task.
4.	The evaluation criteria for both Country and Severity shift are different – AUC vs Accuracy. Is there a reason for this?
5.	The authors should consider adding differences in image pre-processing methods as an additional benchmarking task. While such a task is not purely a naturally occurring phenomenon, it has been shown to be quite even within the Kaggle challenge that the authors mention.
6.	While it is perfectly reasonable that the work uses existing datasets in a structured way, it would be helpful for the authors to comment on ethical considerations and usage of these datasets.

[1] http://www.cvlibs.net/datasets/kitti/
[2] M. T. Le, F. Diehl, T. Brunner and A. Knol, "Uncertainty Estimation for Deep Neural Object Detectors in Safety-Critical Applications," 2018 21st International Conference on Intelligent Transportation Systems (ITSC), 2018, pp. 3873-3878, doi: 10.1109/ITSC.2018.8569637.


**Additional Feedback:**

Nit: Green dot is hard to see on Fig. 4/5

**Documentation:**

The benchmarking experiments are very well documented. The authors should explain the maintenance plan for these models and if future models trained on this benchmark will be released.

**Ethics:**

See 6 in Weaknesses

**Relation To Prior Work:**

The authors appear to do a reasonable job of covering literature related to uncertainty quantification in the DR field, but it would be helpful to talk about other real-world benchmarking datasets for uncertainty and how their analysis is similar and/or different from methods proposed in that work.

**Summary And Contributions:**

This work introduces new benchmarking tasks for evaluating model uncertainty for diabetic retinopathy classification using two existing datasets. Using simulated experimental configurations, it explores how their approach can be used to evaluate performance of Bayesian and non-Bayesian models in cases of distribution shift in data source (e.g. patient population) and label coverage. The work is also accompanied by a well-structured code release that may help simplify interacting with these models.

Overall, the work shows promise but could benefit from more experiments and detail for uncertainty quantification and analysis

---

> ### Author Response · Authors · 2021-09-29
> **Response to Reviewer 7w45 (3/3)**
>
> ### Effect of Preprocessing on Downstream Tasks (cont’d)
>
> We test the downstream performance of MAP estimation (a deterministic model), Deep Ensembles, MC Dropout, and MC Dropout Ensembles on the Country and Severity Shift prediction tasks, varying the `blur_constant` $\in \\{5, 10, 20, 30\\}$.
>
> **Severity Shift (Figure 13, Table 12 in results PDF [here](https://drive.google.com/file/d/1ntutyQ1tN7i8YNxipxIAO_CgqbWvzcqK/view?usp=sharing))**
>
> On the in-domain evaluation dataset, higher `blur_constant` tends to perform better across MAP and MC Dropout, single and ensemble models, and the various referral thresholds.
> However, on the Severity Shift, the MC Dropout variants perform better with lower `blur_constant`.
> This highlights the importance for practitioners to test changes in experimental settings, including preprocessing, across a variety of uncertainty quantification methods.
>
> **Country Shift (Figure 14, Table 13 in results PDF [here](https://drive.google.com/file/d/1ntutyQ1tN7i8YNxipxIAO_CgqbWvzcqK/view?usp=sharing))**
>
> Similarly to the Severity Shift results, higher `blur_constant` tends to perform better on the in-domain evaluation data across methods and referral rates.
> Notably, on the distributionally shifted APTOS data, Deep Ensembles outperform MC Dropout Ensembles, and `blur_constant`$ = 20$ significantly improves performance from the default `blur_constant`$ = 30$ for Deep Ensembles between referral rates 0.4 and 0.7.
> For example, for Deep Ensembles at $\tau = 0.7$, we observe $82.2 \pm 2.5$ AUC with `blur_constant`$ = 20$ versus $67.4 \pm 5.6$ AUC with `blur_constant`$ = 30$.
>
>
> ### Maintenance Plans and Future Models
>
> The benchmarking tasks are part of the [Uncertainty Baselines repository](https://github.com/google/uncertainty-baselines), which is actively maintained. Given the structure of our codebase, extending the benchmarks to new approximate inference methods and updating the network architecture is straightforward. We are currently in the process of adding results for a state-of-the-art computer vision architecture---with over 50M parameters, pretrained on 10M+ images---for a subset of approximate inference methods and hope to be able to release the models soon.
>
>
> ### AUC vs. Accuracy in Country and Severity Shift Evaluation Criteria
>
> Thank you for noting and mentioning this point. We display different evaluation metrics for Country and Severity Shift in the main text because of the following. We consider AUC to be the standard metric for binary classification, and therefore default to this for the Country Shift. However, the evaluation dataset for the Severity Shift is composed entirely of examples which should be classified as “positive”, meaning that the AUC metric is invalid (i.e., the False Positive Rate is undefined). Therefore, we default to reporting selective prediction with respect to accuracy for the distributionally shifted Severity Shift dataset. We also provide both scalar metrics and selective prediction plots for _accuracy_ on the Country Shift (since accuracy is valid there) in Appendix D.
>
>
> ### Ethical Considerations
>
> While the datasets were originally linked to healthcare records, both datasets are fully anonymized and publicly available. One important consideration of our work is that it seeks to highlight the shortcomings of existing methods, and even methods that succeed on the prediction tasks proposed in our submission should be used with caution, as other safety-critical real-world settings may present challenges not reflected in the benchmarking tasks. We have added a discussion of this point to the conclusion (Section 7).
>
>
> ### Uncertainty Quantification Metric
>
> We were unsure what was meant by “details regarding how the uncertainty metric is quantified is unclear in the main text and appendix.”
> Would you mind clarifying whether you are referring to predictive uncertainty estimation, as described in Section 4 (“Uncertainty Estimation”), or to the selective prediction task (which incorporates model predictive uncertainty estimates), which we describe in Section 2.4? Any clarification on what was unclear would be greatly appreciated so that we can revise the relevant sections of the submission accordingly.
>
>
> #​​## Figure 4 and 5 Green Dot Formatting
>
> Thank you for the pointer. We have switched to a more easily visible large black dot.
>
> --
>
> We hope that our response and the new experiments have addressed your questions and concerns. Please let us know if you have any remaining questions.
>
> --
>
> **References**
>
> [1] http://www.cvlibs.net/datasets/kitti.
>
> [2] M. T. Le, F. Diehl, T. Brunner and A. Knol, "Uncertainty Estimation for Deep Neural Object Detectors in Safety-Critical Applications," 2018 21st International Conference on Intelligent Transportation Systems (ITSC), 2018.
>
> [3] C. Leibig, V. Allken, M. Seçkin Ayhan, P. Berens, and S. Wahl, “Leveraging uncertainty information from deep neural networks for disease detection.” Nature Scientific Reports, 2017.

---

> > ### Comment · Reviewer_7w45 · 2021-09-30
> > **Reviewer Response**
> >
> > Thank you for your detailed response. Based on this response, I have adjusted my score to a 7.
> >
> > Regarding uncertainty details, the notation added to 2.4 clarifies the method sufficiently.

---

> ### Author Response · Authors · 2021-09-29
> **Response to Reviewer 7w45 (2/3)**
>
> ### Multi-Class vs. Binary Classification (cont’d)
>
> Below, we describe highlights of the clinical label--binned predictive entropy results which are accessible [here](https://drive.google.com/drive/folders/1j8xAlY-EETi27WziV49j4A29vfMWzwy7?usp=sharing).
> We focus the discussion on single models on the shifted datasets, though we also include evaluation of ensembles and in-domain datasets in the above folder and the revised Appendix D. In each figure, all histograms are normalized and are displayed with the same range on the $x$- and $y$-axis. Some bars of the histograms are cut off because the plots are zoomed-in along the $y$-axis, in order to improve visibility of the important portion.
>
> **Severity Shift**
>
> On the Severity Shift (`severity.pdf` in the folder), we find that MAP, Rank-1, and mean-field variational inference (MFVI) exhibit poor uncertainty estimates on the labels which constitute the distributionally shifted data (labels 3, and 4, with labels appearing on the right of the figure); this can be seen by many of the incorrect predictions also being assigned low predictive uncertainty, i.e., the red distribution having much of its mass near 0.
> This corresponds to a particularly dangerous situation in a real-world deployment: many false negatives with low uncertainty, meaning a specialist might not be able to correct the model’s misdiagnosis.
> MC Dropout, function-space variational inference (FSVI), and radial mean-field variational inference (Radial-MFVI) generate better uncertainty estimates at most severity labels. However, all methods other than MC Dropout perform poorly on label 2 (moderate diabetic retinopathy), either assigning low uncertainty to many incorrect predictions or high uncertainty to many correct predictions.
>
> **Country Shift**
>
> On the Country Shift (`aptos_ood.pdf` in the folder), MAP, Rank-1, and mean-field variational inference (MFVI) have poor predictive uncertainty at all severity levels. MC Dropout and function-space variational inference (FSVI) generate decent predictive uncertainty estimates at all levels except severity 1, where surprisingly, examples that are correctly classified have consistently higher predictive uncertainty than incorrect examples for both methods.
>
>
> ### Effect of Preprocessing on Downstream Tasks
>
> You suggested adding “differences in image pre-processing methods as an additional benchmarking task,” since preprocessing was found to be important in the original EyePACS Kaggle challenge.
> We agree that the effect of changes in preprocessing on performance and uncertainty quantification is an interesting perspective to consider, as we are not aware of any work that has investigated the effect of preprocessing on downstream real-world uncertainty quantification tasks. Below, we describe a set of experiments we conducted to investigate this effect.
>
> In the experiments already included in the submission (Appendix D), we used the preprocessing procedure of the Kaggle competition winner which involved
>
> 1. rescaling the images such that the retinas have a radius of 300 pixels,
>
> 2. subtracting the local average color, computed using Gaussian blur, and finally,
>
> 3. clipping the images to 90% size to remove “boundary effects” [1].
>
> While (1) and (3) are (fairly) standard techniques used to make the data more amenable for use in non-convex optimization, the standard deviation hyperparameter of the Gaussian blur kernel in (2) presupposes some amount of expert knowledge, as the size of the standard deviation governs how visible certain visual artifacts are.
> As such, varying it has a dramatic visual effect on the preprocessed image, and likely required significant tuning.
> (See below link for sample images.)
> In the preprocessing procedure, the standard deviation of the kernel is computed as `target_radius / blur_constant`, where by default, `target_radius=300` and `blur_contant=30`.
> Decreasing the `blur_constant` results in a larger kernel standard deviation, and hence the local average color at each pixel location is computed using a larger window.
> This ultimately results in the preservation of more signal as well as more noise in the input image (because lower-frequency patterns are subtracted).
> See an example [here](https://drive.google.com/file/d/1vc3STfa9wquKJ1v6HEfjK9UrFWuiXmBU/view?usp=sharing) of unprocessed images, along with processed images with the various blur constants.

---

> ### Author Response · Authors · 2021-09-29
> **Response to Reviewer 7w45 (1/3)**
>
> Dear Reviewer 7w45,
>
> Thanks for your feedback. We address general comments in the reply above. Specific comments are addressed below.
>
> --
>
> ### Suggested Related Work
>
> Thank you for suggesting the highly-relevant references [1, 2]. We did not intend to suggest that no other benchmarks exist for real-world tasks and have revised the statements in question.
>
> We have extended our discussion of related work (Section 3) to include the related works you suggested as well as other relevant references.
>
> **Summary of the differences between our work and [1] and [2]**
>
> - [1] is a real-world computer vision benchmark in the autonomous driving domain but does not evaluate the quality of models’ predictive uncertainty estimates.
>
> - [2] uses the dataset from [1] to benchmark the quality of predictive uncertainty estimates but has several limitations: It only considers two methods (as opposed to a large set of state-of-the-art uncertainty quantification methods, as in our submission), and---crucially---each method considers only aleatoric uncertainty. Importantly, [2] does not consider any tasks with distributionally shifted evaluation data, the main challenge to reliable deployment of machine learning models in safety-critical settings. While on the _in-domain_ KITTI evaluation dataset, aleatoric uncertainty estimation may be sufficient, because the evaluation distribution is identical to the training distribution, this may not be the case on distributionally shifted data.
> In contrast, our submission assesses a wide range of methods using both aleatoric and epistemic uncertainty estimates in the context of real-world distribution shifts. We illustrate the importance of epistemic uncertainty at L.371 (Section 6.3) on the Country Shift task; methods which incorporate both aleatoric and epistemic uncertainty significantly outperform the deterministic neural network, which only represents aleatoric uncertainty.
>
>
> ### Multi-Class vs. Binary Classification
>
> We decided to consider binary as opposed to multi-class classification for two reasons:
>
> - First, automated diagnosis pipelines in diabetic retinopathy are mostly concerned with binary decision-making; specifically, whether the patient should be referred to a specialist who would end up re-examining the patient’s retinas. This is described in [3], and was reflected in our correspondence with a domain expert.
>
> - Second, and perhaps more importantly, mild examples (clinical label 1) are very hard to classify, which we confirmed empirically.
>
> We agree that the multi-class problem is relevant and interesting, but since investigating the multi-class prediction problem would require changing all models, uncertainty estimation utilities, and retraining, instead, we conducted a qualitatively similar analysis of predictive uncertainty for correct and incorrect predictions, segmented by non-binarized severity labels 0 to 4 for both the Country and Severity Shifts, where 0 is No Diabetic Retinopathy, and 4 is Proliferative Diabetic Retinopathy.
>
> For each task (Country or Severity Shift) and each method, we bin examples by their underlying clinical label. Then, for each (model, clinical label) pair, we plot the distribution of predictive uncertainty estimates (as measured by the models’ predictive entropy) for correctly and incorrectly predicted examples (in blue and red, respectively).
>
> A model that generates good uncertainty estimates should successfully assign low predictive uncertainty to examples that it classifies correctly (the blue distribution should have most of its mass near 0) and a high predictive uncertainty to incorrect examples (the red distribution should have its mass concentrated at a higher $x$-value).

---

### Author Response · Authors · 2021-09-29
**Response to all reviewers**

### General Comment

We thank the reviewers for their hard work and detailed feedback.

We were glad to see that the reviewers felt that the benchmark “tackles an important problem… which is critical for safety-critical applications” [7w45], has “important social implications, since it aims to evaluate how a medical diagnosis system would perform in a real-world clinical setting” [LzQ9], and presents “tasks [that] are similar to what a real-life predictive model may encounter” [9xr4].

Furthermore, we were pleased that reviewers found that our experimental evaluation “is done appropriately” [7w45], considers a “multitude of uncertainty methods” with “sound evaluation methods and metrics” [9xr4], and successfully “demonstrate[s] that [our] proposed tasks are more informative of real-world model performance than standard metrics” [LzQ9].

Lastly, we were glad that reviewers appreciated our efforts on the code release, noting that the “code release appears to be quite polished and should help simplify user interaction and future experimentation” and is “very well documented” [7w45], and that “all data and code used in the paper is publicly available” [9xr4].


### Suggested Experiments

We conducted two additional experiments suggested by Reviewer 7w45:

1. An investigation of how changes in preprocessing affect downstream predictive performance and quality of uncertainty estimates;

2. An investigation of how the quality of uncertainty estimates varies with the underlying clinical labels 0-to-4.

**Summary of Results**

1.  [Link to Results](https://drive.google.com/drive/folders/1GkQBOC_Y7Jvah_5nK6LVr_U8O8lC5M7K?usp=sharing) |
We varied a Gaussian blur hyperparameter, where a higher “blur constant” results in stronger smoothing on examples during preprocessing (`example.pdf` in folder). We found that on in-domain evaluation across the Severity and Country Shift tasks, the highest blur constant (the default, 30, used throughout previous experiments) performed best. Surprisingly, the trend was reversed for the top performing MC Dropout methods on the Severity Shift distributionally shifted evaluation dataset, with the lowest smoothing improving accuracy at 0 referral by 10% over the default setting. Additionally, for MAP (a deterministic model) and Deep Ensembles on the Country Shift, a middle-range blur constant significantly improves performance over that of the default across referral rates. This experiment highlights the importance for practitioners to test changes in experimental setup across uncertainty quantification methods and metrics.

2. [Link to Results](https://drive.google.com/drive/folders/1j8xAlY-EETi27WziV49j4A29vfMWzwy7?usp=sharing) | We find that MC Dropout and function-space variational inference (FSVI) have the highest quality of uncertainty estimates on the distributionally shifted evaluation datasets, across all clinical severity labels, though no model produces good uncertainty estimates on APTOS label 1.

We have updated the manuscript to reflect your suggestions and updated the supplement (Appendix D) with figures and tables presenting the results above as well as further analysis.

---

### Decision · Program_Chairs · 2021-10-09

**Decision:**

Accept

**Comment:**

The reviewers all liked the paper. The authors' response clarified some important points. In view of that, the authors are strongly invited to take the feedback on board for the final version.